# Pharmacological treatment for connective tissue disease-associated interstitial lung involvement: *Protocol for an overview of systematic reviews and meta-analyses*

**Fotini B. Karassa**[1], **Konstantinos I. Bougioukas**[2], **Eleftherios Pelechas**[1], **Anastasia Skalkou**[3], **Evangelia Argyriou**[4], **Anna-Bettina Haidich**[2] *

1 Division of Rheumatology, Department of Internal Medicine, School of Medicine, Faculty of Health Sciences, University of Ioannina, Ioannina, Greece, 2 Department of Hygiene, Social-Preventive Medicine & Medical Statistics, School of Medicine, Faculty of Health Sciences, Aristotle University of Thessaloniki, Thessaloniki, Greece, 3 Division of Rheumatology, Police Medical Center of Thessaloniki, Thessaloniki, Greece, 4 Rheumatology Unit, Sismanogleio General Hospital, Athens, Greece

* haidich@auth.gr

**Data Availability Statement:** The protocol does not report results. No datasets were generated or analyzed during the current study.

## Abstract

### Background

Interstitial lung disease (ILD) is the most important pulmonary manifestation of connective tissue diseases (CTDs) since it is associated with high morbidity and mortality. However, there is uncertainty on what constitutes the optimal treatment options from a variety of competing interventions. The aim of the overview is to summarize existing evidence of the effectiveness and harm of pharmacological therapies for adults with CTD-ILD.

### Methods

A literature search will be conducted in MEDLINE, the Cochrane Database of Systematic Reviews, DARE, the Centre for Reviews and Dissemination Health Technology Assessment database, Epistemonikos.org, KSR Evidence, and PROSPERO. We will search for systematic reviews with or without meta-analysis that examine pharmacological treatment for CTD-ILD. Updated supplemental search will also be undertaken to identify additional randomized controlled trials. The primary outcomes will be changes in lung function measures and adverse events. The methodological quality of the included reviews will be assessed using the AMSTAR 2 tool. The overall quality of the evidence will be evaluated using the GRADE rating. Summarized outcome data extracted from systematic reviews will be described in narrative form or in tables. For each meta-analysis we will estimate the summary effect size by use of random-effects and fixed-effects models with 95% confidence intervals, the between-study heterogeneity expressed by $I^2$, and the 95% prediction interval. If feasible, given sufficient data, network meta-analysis will be conducted to combine direct and indirect evidence of class and agent comparisons.

**Funding:** The author(s) received no specific funding for this work.

**Competing interests:** The authors have declared that no competing interests exist.

## Discussion

While many factors are crucial in selecting an appropriate treatment for patients with CTD-ILD, evidence for the efficacy and safety of a drug is essential in guiding this decision. Thus, this overview will aid clinicians in balancing the risks versus benefits of the available therapies by providing high-quality evidence to support informed decision-making and may contribute to future guideline development.

## Systematic review registration

MedRxiv: DOI 10.1101/2022.01.25.22269807
 PROSPERO: CRD42022303180

## Introduction

### Rationale

Connective tissue diseases (CTDs) encompass several autoimmune disorders including systemic sclerosis (SSc), rheumatoid arthritis (RA), the inflammatory myopathies, primary Sjogren's syndrome (SS), systemic lupus erythematosus (SLE), and mixed connective tissue disease (MCTD) which can affect any component of the respiratory tract, causing a diverse range of manifestations [1]. Interstitial lung disease (ILD) characterized by inflammation or fibrosis of the pulmonary parenchyma appears to be the most important presentation as it is often progressive and associated with high morbidity and mortality [1, 2]. Consequently, early diagnosis and therapeutic intervention are essential to help prevent worsening of symptoms and decline in pulmonary function. Still, treatment of CTD-associated ILD is a subject of intense debate [3–5] and management of such patients remains difficult despite the rapidly evolving treatment landscape [2, 4, 5]. For many years, immunosuppressive agents were considered the most appropriate drug class for treatment initiation [2, 4, 5]. Yet recently the tyrosine kinase inhibitor nintedanib became the first drug approved by the US Food and Drug Administration (FDA) and the European Medicines Agency (EMA) for the treatment of SSc-ILD [6, 7]. In March 2021, the FDA also approved tocilizumab, an anti-IL-6 receptor humanized monoclonal antibody that blocks IL-6 signaling, for the same indication [8]. Nevertheless, evidence-based guidance on what drug class or individual agent would be optimal as a first-line preference, how to deal with situations in which only weak evidence supports one drug versus another and how to switch to alternate treatment options especially in patients with progressive fibrosing ILDs remains inconclusive [9–13]. The conflicting treatment algorithms [9–13] reflect the variability in management approaches for patients with CTD-ILDs across rheumatologists [14, 15]. In routine clinical practice, physicians must balance a high level of need for treatment in a complex patient group with a potentially progressive disease phenotype against the possibility for adverse events from toxic therapies.

Numerous systematic reviews and meta-analyses on treatment modalities of CTD-related ILD have been published [16–27]. However, there has been no effort to summarize or synthesize the findings of systematic reviews and meta-analyses. Further, safety outcomes have not been adequately studied, the comparative effectiveness of treatments remains uncertain, and there is no clear evidence of relative superiority across the different drug classes or the specific agents [16–27]. Overviews integrating information from systematic reviews and meta-analyses allow a higher-level synthesis of the evidence and better appreciation of the uncertainties and

biases [28, 29]. We therefore plan to conduct the first overview to provide a wider picture on the pharmacological management options for CTD-ILDs that need to be considered and weighed.

## Objective

The objective of this overview is to summarize systematic reviews with or without meta-analysis that assess the effectiveness and harms of pharmacological interventions in patients with CTD-associated ILD. If feasible given sufficient data, network meta-analysis will be conducted to combine direct and indirect evidence of class and agent comparisons with the aim of providing a comprehensive evidence base to inform treatment decisions.

## Methods

### Study design

This protocol conforms to PRISMA-P recommendations [30] (S1 Checklist) and was developed in accordance with current guidelines [31–33]. The reporting of this overview of systematic reviews will be guided by the standards of the Preferred Reporting Items for Overviews of Systematic Reviews including harms checklist (PRIO-harms) [34]. PRISMA extension statement (for the reporting of systematic reviews incorporating network meta-analyses [NMA]) will also be followed, if appropriate [35]. The eligibility criteria for this overview are presented in the PICOS format (Table 1).

### Eligibility criteria

**Population.** *Inclusion criteria.* We will include systematic reviews with or without meta-analysis of randomized controlled trials (RCTs) and/or observational studies of any duration that assess the efficacy or harm of any pharmacological treatment (Tables 1 and 2) in adults

**Table 1. Summary of eligibility criteria for this overview.**

|  | Inclusion criteria | Exclusion criteria |
|---|---|---|
| **Participants** | Adults ≥ 18 years of age with CTD-associated ILD | |
| **Interventions** | Any pharmacological treatment | Non-pharmacological treatments or invasive procedures; nonhuman studies |
| **Comparators** | Another active comparator or placebo | |
| **Outcomes** | | |
| *Primary* | Changes in FVC% and DLCO % predicted | |
| | Number of patients with adverse events (any adverse events, with severe adverse events, with serious adverse events, and with fatal adverse events) | |
| | Number of patients discontinuing treatment due to adverse events | |
| *Secondary* | Survival and mortality (all-cause) | |
| | Dyspnea Index | |
| | Exercise tolerance (6-min walk distance) | |
| | Health-related quality of life | |
| | Change in quantitative HRCT scores | |
| **Study design** | Systematic reviews (without quantitative synthesis) and meta-analyses including RCTs or observational studies; network meta-analyses | Narrative reviews; expert opinions; clinical practice guidelines |
| | RCTs not included in the eligible systematic reviews or meta-analyses | |

**Abbreviations**: CTD: connective tissue disease; DLCO: diffusing capacity of the lung for carbon monoxide; FVC: forced vital capacity; HRCT: high-resolution computed tomography; ILD: interstitial lung disease; RCTs: randomized controlled trials

**Table 2. Intervention classes and individual treatments.**

| Class | Individual treatments |
|---|---|
| DMARDs*/immunosuppressive therapies | Mycophenolate, azathioprine, cyclophosphamide, tacrolimus, methotrexate, leflunomide, corticosteroids, sulfasalazine, hydroxychloroquine, gold/auranofin, ciclosporin, D-penicillamine, tacrolimus, tofacitinib, pomalidomide, iv immunoglobulin |
| Biologic DMARDs** | Tocilizumab, rituximab, abatacept, belimumab, anti-TNF agents [adalimumab, etanercept, infliximab, golimumab, certolizumab pegol], CAT-192, inebilizumab |
| Tyrosine kinase inhibitors/antifibrotic agents | Nintedanib, imatinib mesylate, dasatinib, pirfenidone, SAR100842 |
| Other pharmacological treatments | Bosentan, riocigulat, dabigatran |

**Abbreviations**: DMARDs: disease-modifying antirheumatic drugs; iv: intravenous

*either conventional or targeted synthetic disease-modifying antirheumatic drugs

**including biosimilars

($\geq$ 18 years) with CTD-ILDs. We will consider systematic reviews that recruited patients diagnosed with CTDs mostly associated with ILD. Specifically, systematic reviews on SSc, RA, SS, SLE, MCTD, and the inflammatory myopathies (polymyositis or dermatomyositis) will be eligible for inclusion and only if they used standardized criteria for the definition of the specific CTD as also for the diagnosis of ILDs. A detailed description of the design of eligible publications is provided below in the "Study designs" section.

*Exclusion criteria*. Systematic reviews with or without meta-analysis of patients with ILDs in the context of $\geq$ 1 clinical or serological CTD features without meeting diagnostic criteria or those that included subjects with ILD related to other immune-mediated disorders will not be considered as eligible. Systematic reviews assessing non-pharmacological treatments or invasive procedures, and nonhuman studies will also be excluded.

**Interventions.** A list describing the active agents that have been evaluated for CTD-ILDs is presented in Table 2. We will include publications regardless of whether pharmacological treatment was administered as monotherapy or in combination. Medications may be fixed or flexibly dosed. We will consider any mode of administration.

*Classification of interventions*. Tested pharmacological treatments have been grouped into four intervention classes (Table 2) based primarily on their mechanisms of action [2, 4, 36, 37]. These classes include disease-modifying antirheumatic drugs (DMARDs)/immunosuppressive therapies, biologic DMARDs, tyrosine kinase inhibitors/antifibrotic agents, and other pharmacological treatments. The last category consists of agents that cannot be incorporated elsewhere based on their molecular targets. This classification is in accordance with previously proposed categories of individual treatments investigated across clinical distinct CTDs [4, 23, 38, 39]. Nevertheless, the impact of individual therapies in CTD-ILD seems to be more complex and not solely limited to targeted immunomodulatory or profibrotic pathways [36, 37, 40]. This is especially true for certain tyrosine kinase inhibitors which have shown inconsistent results in the treatment of ILDs despite having partially overlapping inhibition profiles [41]. Hence, the treatment categories may not necessarily reflect the clinically relevant effects of the individual drugs. If NMA will be considered feasible given sufficient data, then grouping individual therapies into meaningful classes is expected to maximize statistical power. In this case, we plan to use the appropriate random-effects NMA model [31, 42] as described in the "Data synthesis and analysis" section.

Some investigational drugs which are being evaluated in ongoing trials [36, 37, 40] may have not been integrated in the four categories since the eligible studies for this overview is

expected to focus largely on published articles. Therefore, we plan to update the included reviews by searches for additional primary studies, as described in the "Study designs" section.

**Comparator.** A different active comparator or placebo.

**Outcomes.** Studies that include the following outcomes will be considered:

*Primary outcomes.*

- *Efficacy (continuous outcome expressed as mean ± standard deviation [SD])*
  Changes from baseline in forced vital capacity (FVC) and diffusing capacity of the lung for carbon monoxide (DLCO) as percentages of the predicted value.

- *Safety*
  The proportion of participants with at least one adverse event (any adverse events, with severe adverse events, with serious adverse events, and with fatal adverse events).

- *Safety*
  The proportion of participants discontinuing treatment due to adverse events.

We intend to categorize adverse events according to the classification outlined in the Medical Dictionary for Regulatory Activities (MedDRA) coding (https://www.meddra.org/) and previous studies [43]. If data will be inadequately reported according to this categorization, then the primary investigators will be conducted to obtain the missing results. In case of no response, we will rely on the summarized information provided in the eligible systematic reviews.

Secondary outcomes.

- *Efficacy (dichotomous outcome)*
  Survival and mortality (all-cause).

- *Efficacy (continuous outcome expressed as mean ± SD)*
  Change in dyspnea index scores as measured by validated questionnaires and assessed with an established rating scale.

- *Efficacy (continuous outcome expressed as mean ± SD)*
  Changes in exercise tolerance using the reproducible 6-min walk distance test.

- *Efficacy (continuous outcome expressed as mean ± SD)*
  Change in health-related quality of life scores as measured by validated questionnaires and assessed with an established rating scale.

- *Efficacy (continuous outcome expressed as mean ± SD)*
  Change in quantitative scores using serial volumetric high-resolution computed tomography (HRCT) scans.

**Study designs.** Articles will be eligible for this overview if the authors had used an explicit, systematic, and reproducible methodology to assemble and synthesize findings of studies that addressed a clearly formulated question [44, 45]. Systematic reviews with or without meta-analysis that included RCTs and/or observational studies (prospective/cohort or retrospective/case-control) will be considered eligible. NMA will also be considered eligible. Narrative reviews, expert opinions, and clinical practice guidelines will be excluded (Table 1).

Considering the rapidly evolving treatment spectrum of CTD-hassociated ILD [2, 4, 5, 36, 37] and the evidence showing that a substantial proportion of published reviews are out of date even one year after their publication [46], we plan to update the included meta-analyses by searches for additional eligible primary studies. The eligibility criteria of this overview

regarding the patient population, interventions, comparators, and outcomes (Tables 1 and 2) will be used to identify only additional RCTs since observational studies are prone to selection bias and confounding [31].

**Language.**   No language restrictions will be applied in the selection of eligible studies.

## Data sources

Pertinent published systematic reviews with or without meta-analysis will be identified through various sources:

- Bibliographic databases and registries of systematic reviews

  We will search the following databases from inception to January 31, 2022:

  ◦ MEDLINE

  ◦ The Cochrane Database of Systematic Reviews (CDSR)

  ◦ The Database of Abstracts of Reviews of Effects (DARE)

  ◦ The Centre for Reviews and Dissemination (CRD) Health Technology Assessment (HTA) database

  ◦ Epistemonikos.org

  ◦ KSR Evidence

  ◦ PROSPERO (International Prospective Register of Systematic Reviews)

- The reference lists of the selected articles will be manually searched.

- Primary investigators will be conducted to obtain additional data that may be missing from the published articles.

  After selecting eligible systematic reviews with or without meta-analysis for this overview based on the predefined criteria (Table 1), an updated supplemental search for recently published RCTs will be done in the following sources:

- Bibliographic databases

  We will search the following databases starting from the last search date of the latest included meta-analysis:

  ◦ MEDLINE

  ◦ The Cochrane Central Register of Controlled Trials (CENTRAL)

- Searches for unpublished and ongoing RCTs will also be undertaken in the following trial registers:

  ◦ The WHO International Clinical Trials Registry Platform (ICTRP). The ICTRP platform receives RCTs from all major trial registries, including ClinicalTrials.gov and the European Clinical Trials Register (EU-CTR).

  ◦ The ClinicalTrials.gov platform will also be searched to retrieve trials that may have not yet been added to the ICTRP.

- The search will be complemented with the perusal of abstracts from the two major rheumatology scientific meetings carried out in the last two years (2020–2021):

◦ The Annual European Congress of Rheumatology (https://www.congress.eular.org/abstract_archive.cfm).

◦ The Annual Meeting of the American College of Rheumatology (https://www.rheumatology.org/Learning-Center/Publications-Communications/Abstract-Archives).

## Search strategy

Two researchers will independently search the databases for relevant systematic reviews with or without meta-analysis. The search strategy was informed by PICOS criteria (Table 1) and will be comprised of three groups of terms relating to systematic reviews [47, 48], CTD-ILDs [1, 4, 5, 10], and interventions [4, 9, 10, 23, 36–40].

Medical subject heading (MeSH) terms and free-text keywords in titles and abstracts that will be used in the initial search will include: "*Lung Diseases, Interstitial*" OR "*Diffuse Parenchymal Lung Disease*", "*Interstitial Lung Diseases*", OR "*Diffuse Parenchymal Lung Diseases*", OR "*Interstitial Lung Disease*", OR "*Lung Disease, Interstitial*" OR "*Pneumonia, Interstitial*" OR "*Interstitial Pneumonia*" OR "*Interstitial Pneumonias*" OR "*Pneumonias, Interstitial*" OR "*Pneumonitis, Interstitial*" OR" *Interstitial Pneumonitides*", OR "*Interstitial Pneumonitis*" OR "*Pneumonitides, Interstitial*". These terms will be combined with highly sensitive search filters for systematic reviews (*#3 "systematic review"[tiab], #4 meta-analysis[pt], #5 intervention\*[ti], #3 OR #4 OR #5*) validated for several databases [47, 48]. Next, the search will combine terms related to specific CTDs ("*connective tissue diseases*", OR "*systemic sclerosis*", OR *scleroderma*, OR "*rheumatoid arthritis*", OR "*inflammatory myopathies*", OR *polymyositis*, OR *dermatomyositis*, OR "*Sjogren's syndrome*", OR *Sjogren*, OR "*systemic lupus erythematosus*", OR *lupus*, OR "*mixed connective tissue disease*") with search filters for systematic reviews (*#1"systematic review"[tiab], #2 meta-analysis[pt], #3 intervention\*[ti], #1 OR #2 OR #3 OR #4*) [47, 48].

We will retrieve additional pertinent published RCTs using combinations of terms related to ILDs (as described above) with highly sensitive search filters for RCTs (*(randomized controlled trial[pt]) OR (controlled clinical trial[pt]) OR (randomized[tiab] OR randomised[tiab]) OR (placebo[tiab]) OR (drug therapy[sh]) OR (randomly[tiab]) OR (trial[tiab]) OR (groups [tiab])) NOT ("animals"[mh] NOT ("animals"[mh] AND "humans"[mh]))* [31]. Finally, the terms for specific autoimmune diseases (as described above) will be appended to the list of specific interventions used in CTD-associated ILD (Table 2).

## Study selection

Two reviewers will screen the retrieved records independently, examine full-text articles, and check inclusion criteria. Firstly, the title and abstract of each of the retrieved citations will be assessed and then potentially eligible articles will be selected for perusal in full text. The online software Rayyan [49] will be used to facilitate first stage screening. Disagreements in the process of selection will be resolved by discussion with a third investigator.

## Management of potentially overlapping systematic reviews

Overlapping of systematic reviews included in overviews stemming from the inclusion of identical primary studies is often underreported and may introduce bias [31]. When faced with overlapping reviews of the same drug in the same patient population, and for the same outcome, we will initially include all relevant publications. Next, we will assess the primary study overlap among all eligible systematic reviews/meta-analyses by producing a citation matrix and calculating the corrected covered area as described in the "Mapping the extent of primary study overlap" section. Should high or very high overlap be detected, we will apply predefined

decision rules to include only some of these systematic reviews and meta-analyses as described below.

## Mapping the extent of primary study overlap

After the selection process, the list of pertinent publications will be carefully reviewed for primary study overlap to avoid double-counting outcome data. To manage overlapping systematic reviews with or without meta-analysis, we will create a citation matrix presenting all the included reviews and their primary studies [50]. Then, the corrected covered area (CCA) will be calculated which provides a numerical measure of the extent of primary study overlap across eligible systematic reviews and meta-analyses [50]. Pairwise CCA as well as CCA for each primary outcome will also be determined [50] and the proposed graphical techniques will be used [51]. Since CCA is not influenced by large reviews, it is expected to reflect the degree of actual overlap. In case we detect high or very high overlap which is interpreted as CCA equal to or more than 10% [50], we plan to retain the review if it is **(a)** the most comprehensive, **(b)** the most recent, and **(c)** the most methodologically rigorous [31, 52–54] using the AMSTAR 2 tool as described in the "Robustness of findings and risk of bias" section.

## Data extraction

Data will be extracted using standardized data extraction templates to ensure consistency of information and appraisal for each eligible study. Pertinent information will be obtained by one member of the review team and checked for accuracy by a senior member of the review team. If there is missing information on methods, lacking outcome data, or discrepant data (i.e., data from the same primary study that is reported differently across systematic reviews), the corresponding authors will be contacted.

Data that will be recorded for the purposes of this overview from the eligible articles [55] include the following: type of the review (systematic review without quantitative synthesis, meta-analysis, or NMA); first author, journal, year of publication, country, and funding sources; whether there was a protocol and if it was accessible; objective of the review, databases searched, date ranges of databases searched, and eligibility criteria; number and type of primary studies (RCTs, observational or both); total number of participants and characteristics of the patient population (specific CTD diagnosis, age range, proportion of females, disease duration); intervention (dose, mode of administration, concomitant medications) and comparison (drug/dose/route of administration or placebo); duration of follow-up; outcomes that are relevant to this overview (statistical model used for the meta-analysis, summary measures with 95% confidence interval [CI] for each outcome, $p$-value, statistics for heterogeneity assessment, sample size and summary estimate from the largest primary study included in each eligible meta-analysis, additional analyses [e.g., subgroup or sensitivity analysis, meta-regression]); instrument used for quality assessment of the primary studies and rating; whether the Grading of Recommendations Assessment, Development and Evaluation (GRADE) approach [31, 56, 57] was used per outcome and the rating and methods for detecting publication bias. For systematic reviews with no quantitative synthesis, we will also record the authors' concluding remarks on their main findings and the reason why a meta-analysis was not attempted.

If an article presents separate meta-analyses on more than one outcome of interest (such as changes in lung function tests, health-related quality of life ratings, or changes in quantitative HRCT scores), those will be recorded separately. In case we encounter studies that have recruited both eligible and ineligible patients, we will try to obtain data on the eligible subpopulation separately. If the data for the eligible subset are not available from the publication (e.g.,

data on CTD-ILD participants as part of the larger ILD patient population), then the primary investigators will be conducted to obtain the missing results.

If NMA will be considered feasible, then we will also extract information from the individual RCTs evaluated in the included meta-analyses as well as from the retrieved trials after the supplemental updated search. Information extracted will include study identifiers and characteristics; participant characteristics; intervention details; and outcome data.

## Robustness of findings and risk of bias

Two reviewers will independently assess the methodological quality and quality of evidence, and disagreements will be resolved by discussion with a third investigator.

We will assess the quality of all eligible articles using the Assessment of Multiple Systematic Reviews 2 (AMSTAR 2) tool [58] since we expect that some systematic reviews included not only RCTs but also non-randomized studies of pharmacological intervention effects. The AMSTAR 2 ranks the quality of a systematic review according to 16 predefined items without generating an overall score [58].

The GRADE framework will be used to rate the overall quality of the evidence. This approach characterizes the quality of a body of evidence based on study limitations, imprecision, inconsistency, indirectness, and publication bias [31, 56, 57]. Since it may not be directly transferable to overviews of systematic reviews to make consistent assessments, we will additionally use the proposed algorithm which assigns GRADE levels of evidence using a set of concrete rules [59].

Finally, if NMA is thought to be an achievable option, the risk of bias at the level of RCTs for the outcomes of interest will be assessed using the Cochrane Risk of Bias (RoB 2.0) tool [60].

## Data synthesis and analysis

We will provide a descriptive table to summarize findings extracted from the eligible systematic reviews. Specifically, key characteristics of each eligible study including interventions, summarized outcomes, quality assessment, and major conclusions will be presented in tables.

We will re-analyze each eligible meta-analysis using the extracted individual study estimates. To yield unified effect size measures, we will re-calculate the non-standardized continuous outcome as well as weighted mean difference into standardized mean difference with 95% CIs and dichotomous outcomes will be expressed using odds ratios with 95% CIs. We will estimate the summary effect size and its 95% CIs with both fixed-effects and random-effects models [61, 62]. We will also calculate the 95% prediction interval (95% PI) for the summary random-effects estimates which further accounts for between-study heterogeneity. The 95% PI is the range in which we expect the effect of a new observation will be for 95% of similar studies in the future [63, 64]. Between-study heterogeneity will be assessed by the $I^2$ metric which is the ratio of between-study variance to the sum of within-study and between-study variances [65, 66]. $I^2$ varies from 0% to 100% [58] with values $> 50\%$ indicating large heterogeneity. When there are few studies, the 95% CI of $I^2$ estimates can be wide [67]. The regression asymmetry test [68] will be used to assess if there is evidence for small-study effects (i.e., whether small studies inflated effect sizes) [69]. Evidence for small-study effects will be considered a $p$-value $< 0.10$ [70]. Additionally, we will explore whether the summary effect size of the random-effects meta-analysis and the effect of its largest component study (the study with the lowest standard error) are concordant in terms of statistical significance [70]. The excess statistical significance test will also be used which determines whether the observed number of studies with nominally significant results ($p < 0.05$) is larger than their expected number [70, 71].

Subgroup analyses according to primary study design (RCTs and observational studies) will be performed. Sensitivity analyses excluding studies of lower methodological quality will also be conducted. We plan to analyze treatment effects according to specific CTD diagnoses (SSc, RA, SS, SLE, MCTD, and the inflammatory myopathies), given sufficient data. In addition, we will examine whether the summary results of overlapping studies are concordant in terms of direction, magnitude, and significance [72, 73]. We will also explore if many relevant publications would be excluded because of the use of decision rules during the study selection [54].

If NMA will be considered feasible assuming that additional RCTs have been published since the most recent meta-analysis, the appropriate random-effects NMA model will be performed to estimate relative treatment effects based on a synthesis of direct (head-to-head trials) and indirect evidence (where two treatments are compared indirectly via a common comparator) for CTD-ILDs [31, 42]. Data from RCTs included in previously published meta-analyses will be combined with those from trials that will be retrieved from the updated search after removing duplicates. We will use the appropriate model to estimate the relative effects of different treatment classes (e.g., biologic DMARDs, tyrosine kinase inhibitors/antifibrotic agents) and of individual treatments within a class (e.g., tocilizumab, rituximab, abatacept). Sources of possible heterogeneity will be explored if excessive heterogeneity across treatment classes is observed.

The statistical analysis and graphics will be done with R software (Version 4.1.1 or later).

## Discussion

The results from this overview will provide an important evidence base for rheumatologists to inform treatment decisions by a comprehensive assessment of the effectiveness and harm of pharmacological interventions in patients with CTD-ILDs. This will help efforts to develop a precision medicine approach to the treatment of a potentially progressive disease manifestation, which can be used in everyday clinical settings. The lack of updated treatment guidelines and of universally agreed-upon treatment algorithms [9–13] for such patients poses substantial obstacles in terms of improving outcomes and in reducing burden to the health care system. It must be recognized, however, that treatment decisions are multifactorial and individualized. Other factors, such as cost-effectiveness should also be considered in the overall therapeutic approach. Yet, reliable information on the effects and safety of available treatments is fundamental in guiding treatment decisions to improve lung function, with consequent potential to reduce organ-specific morbidity and mortality.

## Supporting information

**S1 Checklist. PRISMA-P 2015 checklist.**
(DOCX)

## Author Contributions

**Conceptualization:** Fotini B. Karassa, Anna-Bettina Haidich.

**Methodology:** Fotini B. Karassa, Konstantinos I. Bougioukas, Eleftherios Pelechas, Anastasia Skalkou, Evangelia Argyriou, Anna-Bettina Haidich.

**Supervision:** Anna-Bettina Haidich.

**Writing – original draft:** Fotini B. Karassa.

**Writing – review & editing:** Fotini B. Karassa, Konstantinos I. Bougioukas, Eleftherios Pelechas, Anastasia Skalkou, Evangelia Argyriou, Anna-Bettina Haidich.

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
