## [Decision Letter · Decision Letter 0]

5 Apr 2022

PONE-D-22-03036

Pharmacological treatment for connective tissue disease-associated interstitial lung involvement:  Protocol for an overview of systematic reviews and meta-analyses

PLOS ONE

Dear Dr. Haidich,

Thank you for submitting your manuscript to PLOS ONE. After careful consideration, we feel that it has merit but does not fully meet PLOS ONE’s publication criteria as it currently stands. Therefore, we invite you to submit a revised version of the manuscript that addresses the points raised during the review process.

The reviewers raised a number of concerns which must be addressed. They felt that the objectives of the study should be better delineated, and that there were issues with the methodological approach, including the calculation of CCA and the use of NMA. The reviewers comments can be viewed in full, below.

We look forward to receiving your revised manuscript.

Kind regards,

Natasha McDonald, PhD

Associate Editor

PLOS ONE

Journal Requirements:

Reviewers' comments:

Reviewer's Responses to Questions

**Comments to the Author**

1. Does the manuscript provide a valid rationale for the proposed study, with clearly identified and justified research questions?

Reviewer #1: Yes

Reviewer #2: Yes

2. Is the protocol technically sound and planned in a manner that will lead to a meaningful outcome and allow testing the stated hypotheses?

Reviewer #1: Yes

Reviewer #2: Yes

3. Is the methodology feasible and described in sufficient detail to allow the work to be replicable?

Reviewer #1: Yes

Reviewer #2: Yes

4. Have the authors described where all data underlying the findings will be made available when the study is complete?

Reviewer #1: Yes

Reviewer #2: Yes

5. Is the manuscript presented in an intelligible fashion and written in standard English?

Reviewer #1: Yes

Reviewer #2: Yes

6. Review Comments to the Author

You may also provide optional suggestions and comments to authors that they might find helpful in planning their study.

Reviewer #1: - you should differentiate between systematic reviews and meta-analyses. There are simply systematic reviews with or without meta-analysis. This should be corrected in the objective.

- Although using MedDRA is a good idea, I wonder whether this can work. In an overview you have to rely on the summarized information in SRs gathered from primary studies

- Similar to that is the specification of using validated instruments. Although highly appreciated you will be hardly able to judge this at the SR level

- 'we plan to update the included meta-analyses by searches for additional eligible primary studies.' Does that mean you will definitely search for additional studies or is this just an option?

- I doubt that you will be able to calculate the CCA at outcome level for the full set of SRs. As far as I am aware this has been described in the literature but never tried in practice. It will be important to report whether this was feasible when publishing the results.

- overall, it would be good to explain under what circumstances you will perform an NMA.

Reviewer #2: The Authors have prepared a methodologically sound protocol which will guide the conducting of an important study.

Some minor comment and suggestions are provided below.

Comments on abstract:

Line 36: In the methods section please rephrase so as you do not imply that “systematic review and meta-analyses” are two different entities. It may be sufficient to mention systematic reviews.

Comments on main text:

Line 87: please replace studies with primary studies to help the reader distinguish your unit of analysis (i.e. syst reviews) from primary studies in these systematic reviews. Please amend throughout the text

Line 88: Please provide the measures of the outcomes that you plan to include and clarify/discriminate the listing of outcomes and the corresponding measures.

Line 210: What is the purpose of using the word “formal”?

Line 315: please place CCA after the explanation of the abbreviation

Line 329-332: This reads as if you retain both overlapping reviews but I understand that you will assess the dates of publication, quality etc and keep the one the satisfies your criteria best. Please rephrase.

Line 746: I assume this is criteria for the primary studies included in the eligible studies which you will include. You may want to clarify this on the title of this section.

Line 296: In your RCT filter you plan to exclude humans as MeSH heading (mh), could you please elaborate on this?

7. PLOS authors have the option to publish the peer review history of their article (what does this mean?). If published, this will include your full peer review and any attached files.

Reviewer #1: **Yes: **Dawid Pieper

Reviewer #2: No

---

## [Author Response · Author response to Decision Letter 0]

14 Apr 2022

A. Reviewer # 1:

1. You should differentiate between systematic reviews and meta-analyses. There are simply systematic reviews with or without meta-analysis. This should be corrected in the objective. 

Reply: As suggested, we have corrected in the objective the sentence as: “The objective of this overview is to summarize systematic reviews with or without meta-analysis….”. For consistency reasons, we have made the same correction in the abstract (Methods section) and throughout the text.

2. Although using MedDRA is a good idea, I wonder whether this can work. In an overview you have to rely on the summarized information in SRs gathered from primary studies. 

Reply: We agree with Reviewer # 1 that, since the unit of data extraction is the systematic review, we will probably have to rely on the summarized information provided in the eligible reviews regarding the occurrence of adverse events. Therefore, we have modified the relevant sentence (page 10) as: “We intend to categorize adverse events according to the classification outlined in the Medical Dictionary for Regulatory Activities (MedDRA) coding (https://www.meddra.org/) and previous studies (43). If data will be inadequately reported according to this categorization, then the primary investigators will be contacted to obtain the missing results. In case of no response, we will rely on the summarized information provided in the eligible systematic reviews”.

3. Similar to that is the specification of using validated instruments. Although highly appreciated you will be hardly able to judge this at the SR level. 

Reply: As above, we agree with Reviewer # 1 for the application of validated instruments. We state in the “Robustness of findings and risk of bias” section (page 19, 2nd paragraph): “Since it may not be directly transferable to overviews of systematic reviews to make consistent assessments, we will additionally use the proposed algorithm which assigns GRADE levels of evidence using a set of concrete rules (59)”. Depending on the additional difficulties we may encounter regarding the application of prespecified instruments within the planned overview, this potential limitation will be acknowledged and discussed in the publication.

4. We plan to update the included meta-analyses by searches for additional eligible primary studies.' Does that mean you will definitely search for additional studies or is this just an option? 

Reply: In the “Study designs” section (page 11, 2nd paragraph), we state: “we plan to update the included meta-analyses by searches for additional eligible primary studies. The eligibility criteria of this overview …. will be used to identify only additional RCTs …”. We assume significant coverage gaps to be found in the eligible systematic reviews such as important therapies may have not be examined. We also expect that these systematic reviews will be out of date, even if recently published. Hence, we will definitely update the included systematic reviews by searches for additional eligible RCTs. For this purpose, we have already specified the databases that will be searched for this supplemental search (page 12), the search strategy for the retrieval of the additional RCTs (page 14), the data that will be extracted from these studies (page 18) as well as the instrument that will be used to assess risk of bias (page 19). 

5. I doubt that you will be able to calculate the CCA at outcome level for the full set of SRs. As far as I am aware this has been described in the literature but never tried in practice. It will be important to report whether this was feasible when publishing the results. 

Reply: We agree with Reviewer # 1 that it may be difficult to calculate CCA for each primary outcome but there is a previous publication (PMC7822342) that has provided the CCA at outcome level and we will try to do the same. However, as recommended when publishing the results, we will mention whether this was feasible or not. 

6. Overall, it would be good to explain under what circumstances you will perform an NMA.

Reply: Since a rather large number of competing treatment options are currently available for the connective tissue disease-associated interstitial lung involvement in published RCTs, NMA may provide advantage in gathering information for comparisons between pairs of therapeutic agents that have never been evaluated within individual trials. Other advantages are the potential for more precise estimates than a single direct or indirect estimate and the estimation of the ranking and hierarchy of therapies for the optimal escalation of treatment options, especially in patients with progressive fibrosing disease phenotype. However, assessments of transitivity and consistency will be integral for ensuring the NMA will be valid. Transitivity will be investigated carefully and will be supplemented with a statistical evaluation of consistency. These assessments will be the key points for performing an NMA. 

B. Reviewer # 2:

B1. Comments on abstract

Line 36: In the methods section please rephrase so as you do not imply that “systematic review and meta-analyses” are two different entities. It may be sufficient to mention systematic reviews.

Reply: As suggested, we have rephrased in the methods section, the relevant sentence as: “We will search for systematic reviews with or without meta-analysis….”. Please also see reply to comment A1 above.

B2. Comments on main text

1. Line 87: please replace studies with primary studies to help the reader distinguish your unit of analysis (i.e. syst reviews) from primary studies in these systematic reviews. Please amend throughout the text. 

Reply: As suggested, we have rephrased the sentence as: “We will include systematic reviews with or without meta-analysis of randomized controlled trials (RCTs) and/or observational studies…” to help the reader distinguish that the unit of our analysis is systematic reviews. To further clarify that the term “primary studies” refers to those included in the systematic reviews, similar corrections have been done throughout the text.

2. Line 88: Please provide the measures of the outcomes that you plan to include and clarify/discriminate the listing of outcomes and the corresponding measures. 

Reply: As suggested, we have provided the measures of outcomes and we have also discriminated the listing of outcomes along with the corresponding measures (pages 9 and 10).

3. Line 210: What is the purpose of using the word “formal”? 

Reply: We have deleted the word “formal” to avoid confusion.

4. Line 315: please place CCA after the explanation of the abbreviation. 

Reply: As suggested, the abbreviation has been placed after the explanation of the corrected covered area (page 16).

5. Line 329-332: This reads as if you retain both overlapping reviews but I understand that you will assess the dates of publication, quality etc and keep the one the satisfies your criteria best. Please rephrase.

Reply: We state at the end of the “Mapping the extent of primary study overlap” section: “In case we detect high or very high overlap which is interpreted as CCA equal to or more than 10% (50), we plan to retain the review if it is (a) the most comprehensive, (b) the most recent, and (c) the most methodologically rigorous (31, 52-54) using the AMSTAR 2 tool ….” section.” In more detail, we intend to apply these criteria only if we detect high or very high overlap across the eligible systematic reviews which is interpreted as CCA ≥10%. In such a case we will only retain the systematic reviews fulfilling the former criteria. 

6. Line 746: I assume this is criteria for the primary studies included in the eligible studies which you will include. You may want to clarify this on the title of this section. 

Reply: We have rephrased the title as: “Summary of eligibility criteria for this overview”.

7. Line 296: In your RCT filter you plan to exclude humans as MeSH heading (mh), could you please elaborate on this?

Reply: To exclude animal studies we have modified our RCT filter as: “……. NOT ("animals"[mh] NOT ("animals"[mh] AND "humans"[mh])).

We have also added the PROSPERO registration number at the 3rd page of the revised manuscript. We will also upload the updated PRISMA-P checklist including the PROSPERO registration number.

---

## [Decision Letter · Decision Letter 1]

19 Jul 2022

Pharmacological treatment for connective tissue disease- associated interstitial lung involvement:

 Protocol for an overview of systematic reviews and meta-analyses

PONE-D-22-03036R1

Dear Dr. Haidich,

We’re pleased to inform you that your manuscript has been judged scientifically suitable for publication and will be formally accepted for publication once it meets all outstanding technical requirements.

Kind regards,

George Vousden

Staff Editor

PLOS ONE

Additional Editor Comments (optional):

Reviewers' comments:

Reviewer's Responses to Questions

**Comments to the Author**

1. Does the manuscript provide a valid rationale for the proposed study, with clearly identified and justified research questions?

Reviewer #2: Yes

2. Is the protocol technically sound and planned in a manner that will lead to a meaningful outcome and allow testing the stated hypotheses?

Reviewer #2: Yes

3. Is the methodology feasible and described in sufficient detail to allow the work to be replicable?

Reviewer #2: Yes

4. Have the authors described where all data underlying the findings will be made available when the study is complete?

Reviewer #2: Yes

5. Is the manuscript presented in an intelligible fashion and written in standard English?

Reviewer #2: Yes

6. Review Comments to the Author

You may also provide optional suggestions and comments to authors that they might find helpful in planning their study.

Reviewer #2: No further comments, I feel that the authors have amended the manuscript sufficiently and is now ready for publication.

7. PLOS authors have the option to publish the peer review history of their article (what does this mean?). If published, this will include your full peer review and any attached files.

Reviewer #2: No

---

## [Editor Report · Acceptance letter]

25 Jul 2022

PONE-D-22-03036R1 

Pharmacological treatment for connective tissue disease-associated interstitial lung involvement: *Protocol for an overview of systematic reviews and meta-analyses*

Dear Dr. Haidich:

I'm pleased to inform you that your manuscript has been deemed suitable for publication in PLOS ONE. Congratulations! Your manuscript is now with our production department. 

Kind regards, 

on behalf of

Dr. George Vousden 

Staff Editor

PLOS ONE